# Towards Differential Handling of Various Blur Regions for Accurate Image Deblurring

## Abstract

Image deblurring aims to restore high-quality images by removing undesired degradation. Although existing methods have yielded promising results, they either overlook the varying degrees of degradation across different regions of the blurred image. In this paper, we propose a differential handling network (DHNet) to perform differential processing for different blur regions. Specifically, we design a Volterra block (VBlock) to incorporate nonlinear characteristics into the deblurring network, enabling it to map complex input-output relationships without relying on nonlinear activation functions. To enable the model to adaptively address varying degradation degrees in blurred regions, we devise the degradation degree recognition expert module (DDRE). This module initially incorporates prior knowledge from a well-trained model to estimate spatially variable blur information. Consequently, the router can map the learned degradation representation and allocate weights to experts according to both the degree of degradation and the size of the regions. Comprehensive experimental results show that DHNet effectively surpasses state-of-the-art (SOTA) methods on both synthetic and real-world datasets. Our code is available at https://anonymous.4open.science/r/DHNet-2E3B/DHNet.py.

## 1 Introduction

Image deblurring seeks to restore high-quality images from these blur versions. Due to the ill-posed nature, traditional method Karaali & Jung (2017) attempts to tackle it by imposing various priors to limit the solution space. However, creating such priors is difficult and often lacks generalizability, making them impractical for real-world applications.

In recent years, a variety of Convolutional Neural Networks (CNNs) Cui et al. (2024; 2023a); Kim et al. (2025); Gao et al. (2024) have been developed for image deblurring. The fundamental component of a CNN is a convolutional layer followed by an activation function. The convolution operation ensures local connectivity and translational invariance, while the activation function adds non-linearity to the network. However, CNNs face inherent limitations, such as local receptive fields and a lack of dependence on input content, which restrict their ability to eliminate long-range blur degradation perturbation. To overcome such limitations, Transformers Rao et al. (2025); Potlapalli et al. (2023); Feng et al. (2023) have been introduced in image deblurring. They leverage the adaptive weights of the self-attention mechanism and excel at capturing global dependencies, demonstrating superior performance compared to CNN-based approaches.

While the aforementioned methods have demonstrated strong performance in image deblurring, they have two key drawbacks: (1) they approximate nonlinear properties by stacking numerous nonlinear activation functions, and (2) they overlook the varying degrees of degradation across different blur regions of the image. As shown in Figure 1, the degradation in a blurred image varies across different regions. The areas highlighted by the green box are more severely blurred than those marked by the red box.

To tackle the first drawback, NAFNet Chen et al. (2022) argues that high complexity is unnecessary and proposes a simpler baseline that employs element-wise multiplication instead of nonlinear activation functions, achieving better results with lower computational resources. However, this approach inevitably sacrifices the ability to map complex input-output relationships, making it difficult

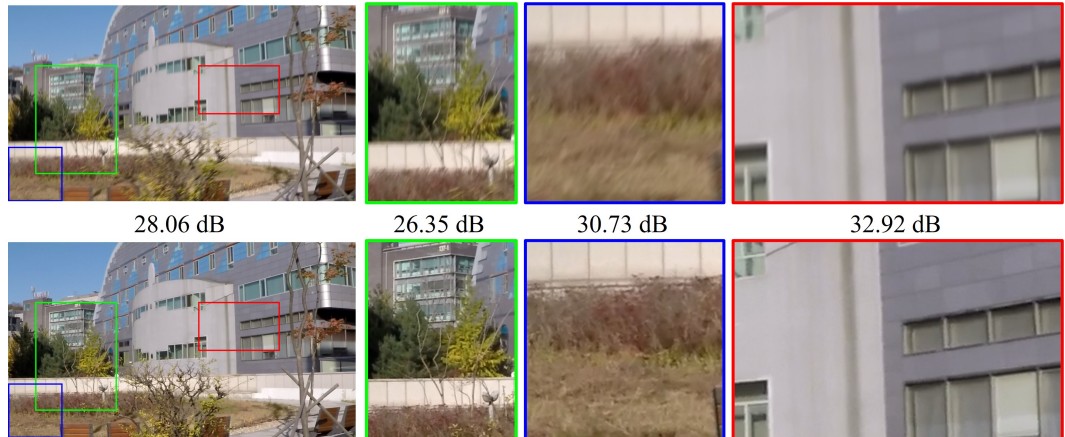

Figure 1: Varying degrees of degradation across different regions. The top row is the blurred image and the bottom row is the sharp image.

to manage more intricate scenes. To address the second issue, AdaRevD Mao et al. (2024) introduces a classifier to assess the degradation degree of image patches. However, this method relies on a limited number of predefined categories and a fixed blur patch size[1] to specify the degree of degradation, which diminishes its effectiveness in adaptively managing different degrees of degradation across various sizes of blurred patches.

Based on the analyses presented above, a natural question arises: Is it feasible to devise a network that effectively performs differential processing for different blur regions with less nonlinear activation function? In pursuit of this objective, we propose DHNet for efficient image deblurring, incorporating several key components. 1) We design a Volterra block (VBlock) to investigate nonlinearity within the network. VBlock avoids using traditional nonlinear activation functions and instead employs Volterra kernel to enhance linear convolution by facilitating interactions between image pixels. This approach approximates non-linearity without the computational overhead associated with stacking numerous nonlinear functions to map complex input-output relationships. 2) To adaptively identify degradation degrees across varying sizes of blur regions, we propose a degradation degree recognition expert module (DDRE). DDRE first integrates prior knowledge from a well-trained model to estimate the spatially variable blur information. This enables the router to map the learned degradation representation and assign weights to experts based on both the degree of degradation and the size of the regions. We conduct extensive experiments to validate the effectiveness of the proposed networks, demonstrating their remarkable performance advantage over state-of-the-art approaches. As illustrated in Figure 2, our DHNet model achieves SOTA performance while requiring less computational cost compared to existing methods.

The main contributions are summarized as follows:

1. We propose an efficient and effective framework for image deblurring, called DHNet, which excels at differentially handling various blur regions while maintaining lower computational costs.

2. We design a Volterra block (VBlock) to investigate non-linearity within the network, avoiding the previous operation of stacking numerous nonlinear functions to map complex input-output relationships.

3. We devise a degradation degree recognition expert module (DDRE), enabling the model adaptively deal with the different degradation degrees of the degraded region.

4. Extensive experiments demonstrate that the proposed DHNet achieves promising performance compared to state-of-the-art methods across synthetic and real-world benchmark datasets.

---

[1] AdaRevD groups the patches into six degradation degrees based on the PSNR between the blurred patch and the sharp patch, and then uses a relatively large patch size of 384 x 384 to classify the degradation degree of each blurred patch.

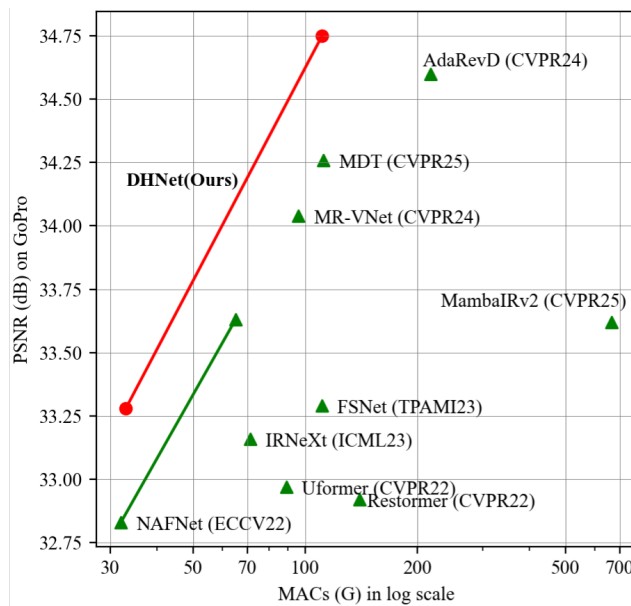

Figure 2: Computational cost vs. PSNR on the GoPro dataset Nah et al. (2016). Our DHNet achieve the SOTA performance with up to 48.9% of cost reduction.

## 2 RELATED WORK

### 2.1 IMAGE DEBLURRING

The ill-posed nature of image deblurring leads many conventional approaches Dong et al. (2011); Karaali & Jung (2017) to rely on hand-crafted priors. Although these priors can assist in blur removal, they often lack generalization.

Rather than manually designing image priors, many methods focus on developing various deep CNNs Cui et al. (2024); Roheda et al. (2024b); Cui et al. (2023b). CGNet Ghasemabadi et al. (2024) integrates a global context extractor to effectively gather global contextual information. IRNeXt Cui et al. (2023a) rethinks CNN design and introduces an efficient network. FSNet Cui et al. (2024) utilizes multi-branch and content-aware modules to dynamically select the most informative components. MR-VNet Roheda et al. (2024b) proposes a novel architecture that leverages Volterra layers for both image and video restoration. ELEDNet Kim et al. (2025) reduces noise while preserving structural details leverages cross-modal information. Nonetheless, the inherent characteristics of convolutional operations limit the models' ability to address long-range degradation disturbances.

To tackle these limitations, Transformers Rao et al. (2025); Zhou et al. (2024) have been utilized in image deblurring. They effectively capture global dependencies through the adaptive weights of the self-attention mechanism, outperforming CNN-based methods. However, the quadratic time complexity of the self-attention mechanism increases the computational burden. To mitigate this issue, Uformer Wang et al. (2022) and U$^2$former Feng et al. (2023) implement self-attention using a window-based approach. In contrast, MRLPFNet Dong et al. (2023), and DeblurDiNAT Liu et al. (2024) compute self-attention across channels rather than in the spatial dimension. FFTformer Kong et al. (2023) leverages frequency domain properties to estimate scaled dot-product attention.

While the aforementioned methods have two main drawbacks: (1) they approximate nonlinear properties by stacking many nonlinear activation functions, and (2) they fail to account for the varying degrees of degradation in different blur regions. Although NAFNet Chen et al. (2022) uses element-wise multiplication instead of nonlinear activation functions, it struggles with more complex scenes. AdaRevD Mao et al. (2024) introduces a classifier to assess the degradation degree of image patches, but it lakes adaptive flexibility. In this paper, we propose a differential handling network to achieve nonlinear properties with fewer nonlinear activation functions while adaptively handling varying degradation degrees in blur regions of different sizes.

## 2.2 VOLTERRA SERIES

The Volterra series is a model for nonlinear behavior that effectively captures "memory" effects. With its capability, Volterra Filters have been applied in deep learning. To enhance non-linearity and complement traditional activation functions,NCF Zoumpourlis et al. (2017) introduces a single layer of Volterra kernel-based convolutions. VNNs Roheda et al. (2024a) proposes a cascaded approach of Volterra Filtering to significantly reduce the number of parameters. VolterraNet Banerjee et al. (2022) presents a novel higher-order Volterra convolutional neural network as samples of functions on Riemannian homogeneous spaces. MR-VNet Roheda et al. (2024b) employs Volterra layers to effectively introduce nonlinearities. However, it captures only second-order nonlinearities while neglecting first-order effects, which can reduce attention to local features and increase the risk of gradient explosion. To address these issues, we design a VBlock that approximates nonlinearities without the computational cost of stacking multiple nonlinear functions.

## 3 METHOD

In this section, we begin with an overview of the DHNet pipeline. Following that, we delve into the details of the differential handling block (DHBlock), which consists of the Volterra block (VBlock) and the degradation degree recognition expert module (DDRE). **More proofs in the Appendix A.**

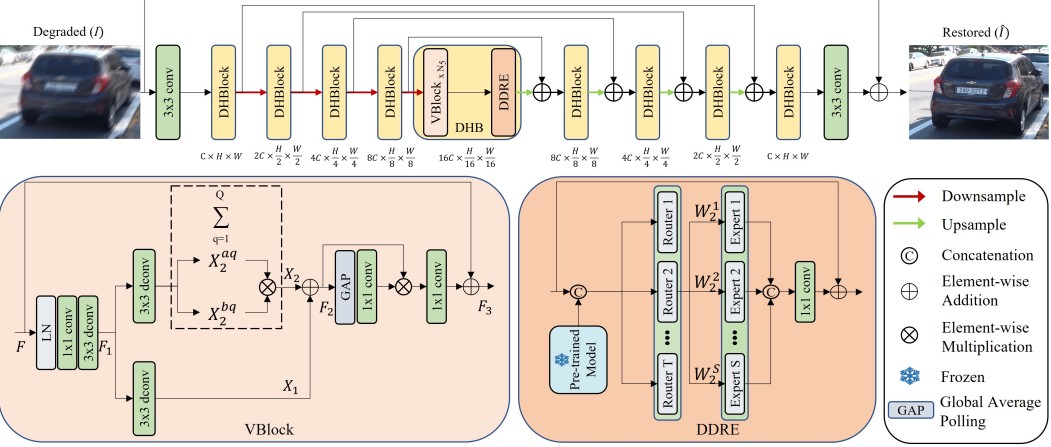

Figure 3: The overall architecture of DHNet mainly consists of the DHBlock, which includes the VBlock and the DDRE. DDRE is shown in the one-router case for clarity.

## 3.1 OVERALL PIPELINE

Our proposed DHNet, depicted in Figure 3, employs a hierarchical encoder-decoder architecture to facilitate effective hierarchical representation learning. Each encoder-decoder includes a DHBlock, which consists of $N_{i \in [1,\dots,9]}$ VBlocks and a degradation degree recognition expert module (DDRE). Given a degraded image $\mathbf{I} \in \mathbb{R}^{H \times W \times 3}$, DHNet first applies convolution to extract shallow features $\mathbf{F_s} \in \mathbb{R}^{H \times W \times C}$ (where $H$, $W$, and $C$ represent the height, width, and number of channels of the feature map, respectively). These shallow features pass through a four-scale encoder sub-network, where the resolution is progressively reduced while the number of channels increases. The resulting in-depth features then move to a middle block, and the deepest features are processed by a four-scale decoder, which gradually restores them to their original size. Finally, we apply convolution to the refined features to generate a residual image $\mathbf{X} \in \mathbb{R}^{H \times W \times 3}$. This residual image is added to the degraded image to produce the restored image: $\hat{\mathbf{I}} = \mathbf{X} + \mathbf{I}$.

## 3.2 DIFFERENTIAL HANDLING BLOCK

While existing image deblurring methods have demonstrated commendable performance, they often rely on stacking numerous nonlinear activation functions to approximate nonlinear properties, and

neglecting the varying degradation degrees across different blurred regions. To address this issue, we design the differential handling block (DHBlock), enabling differential processing of diverse blur regions and utilizing Volterra kernel to capture complex input-output relationships. Formally, given the input features at the $(l-1)_{th}$ block $X_{l-1}$, the procedures of DHBlock can be defined as:

$$X_l^{'} = VBlock_{N_i}(...(VBlock_1(X_{l-1}))...)$$
$$X_l = DDRE(X_l^{'}) \tag{1}$$

where $N_i$ represents the number of VBlocks in the DHBlock, while $X_l^{'}$ and $X_l$ denote the outputs from the $N_i$ Volterra blocks (VBlocks) and the degradation degree recognition expert module (DDRE), which are detailed below.

### 3.3 VOLTERRA BLOCK

The strong performance and versatility of deep learning-based image deblurring models Cui et al. (2024); Mao et al. (2024) can be attributed to their nature as universal approximators. They achieve this by stacking numerous nonlinear activation functions, allowing them to fit any nonlinear function. In order to reduce the system complexity caused by an excess of nonlinear activation functions, we design a Volterra block (VBlock) that utilizes the Volterra kernel to explore non-linearity within the network. Instead of relying on nonlinear activation functions, VBlock employs higher-order convolutions to enhance linear convolution by enabling interactions between image pixels. As shown in Figure 3, unlike MR-VNet Roheda et al. (2024b), which treats the Volterra filter as a complete block, our VBlock incorporates it as part of a block. The VBlock first captures local features before entering the Volterra filter, and then applies two 3x3 convolutions for each order branch. Specifically, our VBlock takes an input tensor $F$, first applies layer normalization (LN), and then encodes channel-wise context to obtain the feature $F_1$ as follows:

$$F_1 = W_d^0 W_p^0 LN(F) \tag{2}$$

where $W_p^{(\cdot)}$ denotes the $1 \times 1$ point-wise convolution, and $W_d^{(\cdot)}$ represents the $3 \times 3$ depth-wise convolution.

The resulting feature $F_1$ is further processed by the second-order Volterra kernel to capture complex input–output relationships through pixel interactions. In this work, the second-order Volterra kernel is computed as a product of traditional correlation operations, allowing approximation with arbitrary precision. Furthermore, the separability assumption of the kernels enables efficient computation, which is particularly beneficial in network settings. The process is as follows:

$$F_2 = X_1 \oplus X_2 = W_d^1 F_1 \oplus \left(\sum_{q=1}^{Q} X_2^{aq} \otimes X_2^{bq}\right) = W_d^1 F_1 \oplus \left(\sum_{q=1}^{Q} W_d^{2aq} F_1 \otimes W_d^{2bq} F_1\right) \tag{3}$$

where $W_d^{2aq}$ and $W_d^{2bq}$ are learnable weight, $Q$ represents the desired rank of approximation.

**Theorem 1.** Any continuous function can be approximated using a Volterra kernel.

**Proof.** Any continuous nonlinear function can be approximated by a polynomial. Using a Taylor expansion at $x_0$, the nonlinear function $\sigma(\cdot)$ can be represented as:

$$\sigma_t(x) = f(x_0) + ... + \frac{f^n(x_0)}{n!}(x - x_0)^n + R_n(x) = \alpha_0 + \alpha_1 x + ... + \alpha_n x^n + R_n(x) \tag{4}$$

For a memoryless higher-order Volterra kernel, it can be formulated as:

$$\sigma_v(x) = h_0 + h_1 x + ... + h_n x^n \tag{5}$$

where $h_n$ represents the $n_{th}$ order weight, which is learned during the training process. The function $\sigma_v(x)$ can be considered an $n_{th}$ order approximation of $\sigma_t(x)$, with the error defined as:

$$error = (\alpha_0 - h_0) + ... + (\alpha_n - h_n)x^n + R_n(x) \tag{6}$$

For a finite polynomial, the absolute error of the expansion can be quantitatively expressed using the Taylor Remainder as follows:

$$|error| \leqslant R_n(x) = \frac{f^{n+1}(\xi)}{(n+1)!}(x - x_0)^{n+1} \tag{7}$$

where $\xi \in (x_0, x)$. Due to the complexity of higher-order Volterra kernels, this paper utilizes a cascade of second-order Volterra kernels to approximate the higher-order behavior.

**Theorem 2.** A Volterra kernel provides a more adaptive representation than activation functions.

**Proof.** For activation functions such as ReLU, sigmoid, and tanh, the coefficients $\alpha_n$ in their Taylor expansions are predetermined[2]. Specifically, a sigmoid activation can be approximated as:

$$\sigma_s(x) = \frac{1}{1 + e^{-x}} = \frac{1}{2} + \frac{1}{4}x - \frac{1}{48}x^3 + ... \tag{8}$$

The expansion coefficient $h_n$ of the Volterra kernel can be continuously adjusted and learned during training, allowing for greater adaptability and more precise mapping of complex scenes. As a result, the feature $F_2$ in Eq. 3 exhibits nonlinear characteristics and offers greater flexibility compared to features obtained through nonlinear activation functions (e.g. sigmoid). Finally, we apply channel attention to $F_2$ to obtain the final output features $F_3$ of VBlock as follows:

$$F_3 = W_p^2(F_2 \otimes (W_p^1 GAP(F_2))) \oplus F \tag{9}$$

### 3.4 DEGRADATION DEGREE RECOGNITION EXPERT MODULE

While existing image deblurring methods Roheda et al. (2024b) have demonstrated strong performance, they overlook the inconsistency in the degradation degree across different regions. As illustrated in Figure 1, the degree of degradation in a blurred image can differ significantly between areas. Thus, it is clearly unreasonable for these methods to assume that all degraded areas share the same degree of degradation. In contrast to AdaRevD Mao et al. (2024), which relies on a limited set of predefined categories and a fixed blur patch size, DDRE first incorporates prior knowledge from a well-trained model to estimate the spatially variable blur information. This enables DDRE to allow the router to map the learned degradation representation and assign weights to experts according to both the degree of degradation and the size of the regions.

Unlike conventional mixture-of-experts approaches Yu et al. (2024); Chen et al. (2023), which typically select only a single router and a subset of experts for each router, our DDRE exploits all routing paths by dynamically assigning weights to every expert along each router. Each expert consists of convolutions with different receptive field sizes, enabling the model to capture degradation patterns across regions of varying scales. For clarity, Figure 3 illustrates the case of a single router. Given an input feature map $F_4$, we first integrate prior knowledge $P$ extracted from a pre-trained model, which is kept frozen during training to reduce memory consumption:

$$F_5 = W_p^3[F_4, P] \tag{10}$$

where $[\cdot]$ denotes concatenation. The aggregated feature map $F_5$, enriched with degradation information, is then fed into each router, which dynamically assigns weights to its experts as follows:

$$F_6 = [W_j^k E_k(F_5)] \tag{11}$$

where $W_j^k$ denotes the learnable weight assigned by router $j \in 1, 2, \ldots, T$ to expert $k \in 1, 2, \ldots, S$, and $E_k(\cdot)$ represents the operations of expert $k$. Finally, the DDRE output is obtained by:

$$F_7 = W_p^4 F_6 \oplus F_4 \tag{12}$$

Through this design, DDRE adaptively captures degradation information at multiple scales and effectively handles varying degrees of degradation across different image regions.

## 4 EXPERIMENTS

In this section, we outline the experimental settings and present both qualitative and quantitative comparisons. We then conduct ablation studies to highlight the effectiveness of our approach.

---

[2]The term "predetermined" refers to a fixed value that remains constant.

Table 1: Quantitative evaluations of the proposed approach against state-of-the-art motion deblurring methods. Our DHNet and DHNet-B are trained only on the GoPro dataset.

| Methods | GoPro PSNR ↑ | GoPro SSIM ↑ | HIDE PSNR ↑ | HIDE SSIM ↑ |
|---|---|---|---|---|
| UFPNet Fang et al. (2023) | 34.06 | 0.968 | 31.74 | 0.947 |
| MambaIR Guo et al. (2024) | 33.21 | 0.962 | 31.01 | 0.939 |
| ALGNet-B Gao et al. (2024) | 34.05 | 0.969 | 31.68 | 0.952 |
| MR-VNet Roheda et al. (2024b) | 34.04 | 0.969 | 31.54 | 0.943 |
| FSNet Cui et al. (2024) | 33.29 | 0.963 | 31.05 | 0.941 |
| AdaRevD-B Mao et al. (2024) | 34.50 | 0.971 | 32.26 | 0.952 |
| XYScanNet Liu et al. (2025) | 33.91 | 0.968 | 31.74 | 0.947 |
| MDT Chen et al. (2025) | 34.26 | 0.969 | 31.84 | 0.948 |
| **DHNet(Ours)** | 33.28 | 0.964 | 31.75 | 0.948 |
| **DHNet-B(Ours)** | **34.75** | **0.973** | **32.37** | **0.953** |

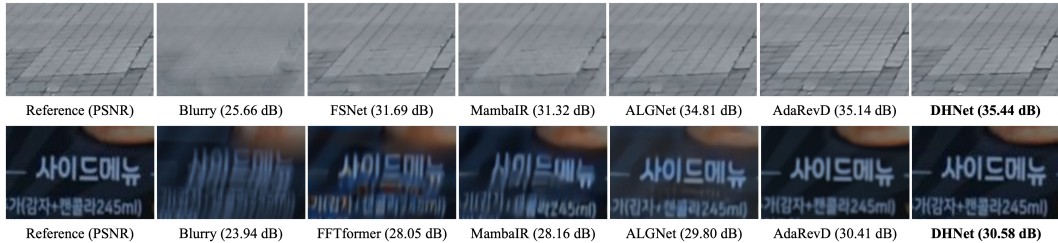

Reference (PSNR)    Blurry (25.66 dB)    FSNet (31.69 dB)    MambaIR (31.32 dB)    ALGNet (34.81 dB)    AdaRevD (35.14 dB)    **DHNet (35.44 dB)**

Reference (PSNR)    Blurry (23.94 dB)    FFTformer (28.05 dB)    MambaIR (28.16 dB)    ALGNet (29.80 dB)    AdaRevD (30.41 dB)    **DHNet (30.58 dB)**

Figure 4: Image deblurring comparisons on the synthetic dataset Nah et al. (2016)(Top) and real-world dataset Rim et al. (2020)(Bottom). Our DHNet recovers image with clearer details.

## 4.1 EXPERIMENTAL SETTINGS

We train separate models for different tasks, and unless otherwise specified, the following parameters are utilized. In our model, $N_{i \in [1,2,3,4,5,6,7,8,9]}$ are set to $\{1, 1, 1, 28, 1, 1, 1, 1, 1\}$. In terms of VBlock, we set $Q = 4$ (Eq. 3). In terms of DDRE, we set $S = 5$ for the number of experts and $T = 4$ for the number of routers. We use the Adam optimizer Kingma & Ba (2014) with parameters $\beta_1 = 0.9$ and $\beta_2 = 0.999$. The initial learning rate is set to $5 \times 10^{-4}$ and gradually reduced to $1 \times 10^{-6}$ using the cosine annealing strategy Loshchilov & Hutter (2016). The batch size is chosen as 32, and patches of size $256 \times 256$ are extracted from training images. Data augmentation includes both horizontal and vertical flips. Given the high complexity of image motion deblurring, we set the number of channels to 32 for DHNet and 64 for DHNet-B. In DHNet, we do not use a pre-trained model in the DDRE, whereas DHNet-B utilizes UFPNet Fang et al. (2023).

## 4.2 EXPERIMENTAL RESULTS

### 4.2.1 EVALUATIONS ON THE SYNTHETIC DATASET

We present the performance of various image deblurring methods on the synthetic GoPro Nah et al. (2016) and HIDE Shen et al. (2019) datasets in Table 1. Overall, our DHNet outperforms competing approaches, yielding higher-quality images with superior PSNR and SSIM values. Specifically, compared to the previous best method, AdaRevD-B Mao et al. (2024), our DHNet-B shows an improvement of 0.25 dB on the GoPro dataset. Notably, although our model was trained exclusively on the GoPro dataset, it still achieves SOTA results (32.37 dB in PSNR) on the HIDE dataset, demonstrating its excellent generalization capability. Furthermore, Figure 2 shows that DHNet not only achieves SOTA performance but also reduces computational costs. The performance continues to improve with an increase in model size, highlighting the scalability of our approach. Figure 4 shows that our model produces more visually pleasing results.

Table 2: Quantitative evaluations on the real-word dataset.

| Methods | RealBlur-R | | RealBlur-J | |
|---|---|---|---|---|
| | PSNR ↑ | SSIM ↑ | PSNR ↑ | SSIM ↑ |
| FFTformer Kong et al. (2023) | 40.11 | 0.973 | 32.62 | 0.932 |
| UFPNet Fang et al. (2023) | 40.61 | 0.974 | 33.35 | 0.934 |
| MambaIR Guo et al. (2024) | 39.92 | 0.972 | 32.44 | 0.928 |
| ALGNet Gao et al. (2024) | 41.16 | 0.981 | 32.94 | 0.946 |
| MR-VNet Roheda et al. (2024b) | 40.23 | 0.977 | 32.71 | 0.941 |
| AdaRevD-B Mao et al. (2024) | 41.09 | 0.978 | 33.84 | 0.943 |
| **DHNet(Ours)** | 41.30 | **0.984** | 33.78 | 0.952 |
| **DHNet-B(Ours)** | **41.33** | 0.983 | **34.28** | **0.953** |

Table 3: Ablation study on individual components.

| Net | VBlock | DDRE | PSNR | △ PSNR |
|---|---|---|---|---|
| (a) | | | 33.62 | - |
| (b) | ✔ | | 34.05 | +0.43 |
| (c) | | ✔ | 34.32 | +0.70 |
| (d) | ✔ | ✔ | 34.75 | +1.13 |

### 4.2.2 EVALUATIONS ON THE REAL-WORLD DATASET

We further assess the performance of our DHNet on real-world images from the RealBlur dataset Rim et al. (2020). As shown in Table 2, compared to the previous best method, AdaRevD-L Mao et al. (2024), our approach achieves improvements of 0.14 dB and 0.32 dB on the RealBlur-R and RealBlur-J datasets, respectively. Figure 4 demonstrates that DHNet produces clearer images with finer details and structures.

## 4.3 ABLATION STUDIES

### 4.3.1 EFFECTS OF INDIVIDUAL COMPONENTS

To evaluate the effectiveness of each module, we use NAFNet Chen et al. (2022) as our baseline model and subsequently replace or add the modules we have designed. As shown in Table 3(a), the baseline achieves 33.62 dB PSNR. Each module combination leads to a corresponding performance improvement. Specifically, replacing NAFBlock Chen et al. (2022) with our VBlock enhances the performance by 0.43 dB (Table 3(b)). Adding the DDRE module can significantly boost the model's performance from 33.62 dB to 34.32 dB (Table 3(c)). When all modules are combined (Table 3(d)), our model achieves a 1.13 dB improvement over the baseline.

### 4.3.2 DESIGN CHOICES FOR VBLOCK

VBlock facilitates the generation of nonlinear interactions through the interactions between image pixels. To evaluate the effectiveness of VBlock, we first use SG Chen et al. (2022) to replace the Volterra kernel and then examine the impact of varying the kernel rank. Table 7 shows the SG Chen et al. (2022) achieves a PSNR of 34.12 dB, and performance improves when we implement our Volterra kernel. We further visualize the feature maps in Figure 5 to highlight the advantages. The results clearly demonstrate that features obtained with the Volterra kernel are more detailed (as indicated by the red box) and better suited for handling complex scenes. Additionally, we analyze

Table 4: Results of alternatives to VBlock.

| Net | SG Chen et al. (2022) | Volterra | Q | PSNR | △ PSNR | MACs(G) |
|---|---|---|---|---|---|---|
| (a) | ✔ | | 0 | 34.12 | - | 106 |
| (b) | | ✔ | 1 | 34.25 | +0.13 | 107 |
| (c) | | ✔ | 2 | 34.42 | +0.30 | 109 |
| (d) | | ✔ | 4 | 34.75 | +0.63 | 111 |
| (e) | | ✔ | 8 | 34.76 | +0.64 | 119 |

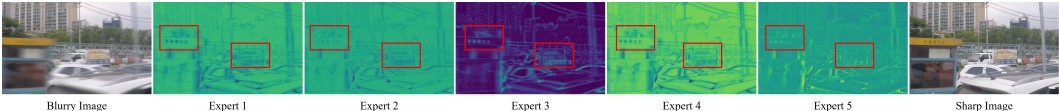

Figure 5: Effect of the proposed VBlock. Zoom in for the best view.

Table 5: Results of alternatives to DDRE, where $S$ and $T$ denote the number of experts and routers.

| Net | Pre-trained NAFNet Chen et al. (2022) | UFPNet Fang et al. (2023) | T | S | PSNR | $\triangle$ PSNR |
|-----|-----|-----|---|---|------|------------------|
| (a) |   |   | 0 | 0 | 34.05 | - |
| (b) |   |   | 4 | 5 | 34.26 | +0.21 |
| (c) | ✔ |   | 4 | 5 | 34.47 | +0.35 |
| (d) |   | ✔ | 4 | 5 | 34.75 | +0.70 |
| (e) |   | ✔ | 4 | 3 | 34.56 | +0.51 |
| (f) |   | ✔ | 4 | 8 | 34.61 | +0.56 |
| (g) |   | ✔ | 2 | 5 | 34.51 | +0.46 |
| (h) |   | ✔ | 6 | 5 | 34.57 | +0.52 |

the effect of the $Q$ value on the performance. As we increase the rank $Q$, performance consistently improves, but this also increases system complexity. To achieve a balance between efficiency and performance, we choose an experimental setting with $Q = 4$.

Figure 6: Effect of the proposed DDRE. Zoom in for the best view.

### 4.3.3 DESIGN CHOICES FOR DDRE

To evaluate the effectiveness of DDRE, we conduct experiments with various model variants, as shown in Table 5. When we introduce experts (Table 5 (b)) into the baseline model (Table 5 (a)), performance improves from 34.05 to 34.26, highlighting the model's capability to address degradation in a differentiated manner. Further enhancement occurs when we incorporate external degradation pattern knowledge, leading to even greater performance gains (Table 5 (c) and (d)). Additionally, we observe that different pre-trained models (NAFNet Chen et al. (2022), UFPNet Fang et al. (2023)) yield varying performance outcomes. We also investigate the impact of the number of routers and experts on the experimental results. Our findings indicate that the combination of $S = 5$ and $T = 4$ (Table 5 (d)) yields the best results. A smaller number (Table 5 (e) (g)) fails to adequately identify the degradation degree. While an excessive number (Table 5 (f) (h)) tends to focus on the severely degraded regions, and the lightly degraded regions will receive relatively less attention, ultimately hindering their restoration. To further demonstrate the effectiveness of our DDRE module, we visualize the features of a routing path weighted over all experts, as shown in Figure 6. It is clear that each expert handles the inconsistent degradation region as well as the degree.

## 5 CONCLUSION

In this paper, we propose a differential handling network (DHNet) aimed at accurately deblurring images by processing different blur regions. Specifically, we design a DHBlock consisting of VBlock and DDRE. The VBlock utilizes the Volterra kernel to explore non-linearity within the network to map complex input-output relationships. Meanwhile, the DDRE integrates prior knowledge from a well-trained model to estimate spatially variable blur information, allowing the router to map the learned degradation representation and assign weights to experts based on the degree of degradation and the size of the regions. Experimental results demonstrate that our DHNet outperforms state-of-the-art approaches.

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

## A  APPENDIX

### A.1  OVERVIEW

## B  MOTIVATION ANALYSIS

Although existing image deblurring methods Kong et al. (2023); Roheda et al. (2024b) have demonstrated strong performance, they fail to account for the varying degrees of degradation across different regions. As shown in Figure 7, the blur intensity differs across regions of the image. Additionally, Figure 8 highlights that larger blurred regions are more challenging to recover. To address the first issue, AdaRevD Mao et al. (2024) introduces a classifier to assess the degradation degree of image patches. However, AdaRevD Mao et al. (2024) relies on a limited set of predefined categories and a fixed blur patch size, which restricts its ability to effectively adapt to different degradation degrees across varying patch sizes. This limitation prevents it from adequately solving the second problem.

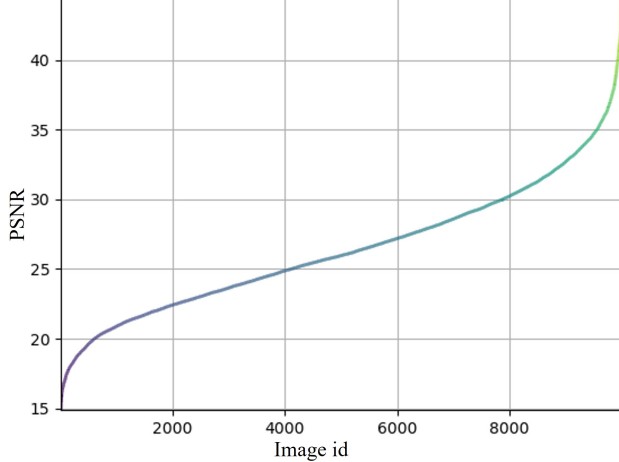

Figure 7: The ranked PSNR curve of the different blur region from GoPro Nah et al. (2016) test set.

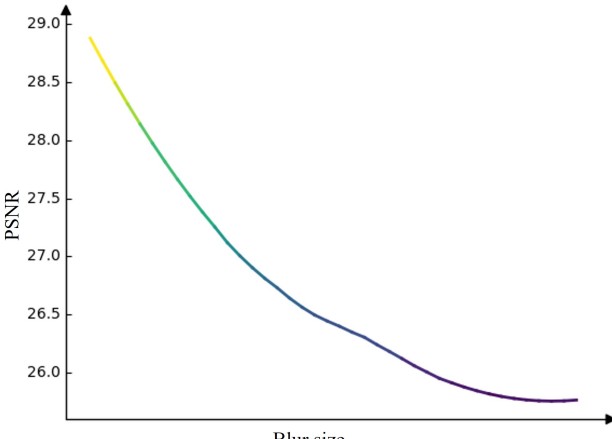

Figure 8: The PSNR curve of the different blur size from GoPro Nah et al. (2016) test set.

Table 6: The configuration of each expert.

| Expert | Configuration |
|--------|---------------|
| (1) | 1x1 depthwise separable convolution |
| (2) | 3x3 depthwise separable convolution |
| (3) | 5x5 depthwise separable convolution |
| (4) | 7x7 depthwise separable convolution |
| (5) | 9x9 depthwise separable convolution |

In this paper, we propose the degradation degree recognition expert (DDRE) module which enables the model to adaptively handle varying degrees of degradation in blurred regions. Unlike conventional mixture of experts methods Yu et al. (2024); Chen et al. (2023), where each router selects only one expert and a subset of experts for processing, our DDRE first integrates prior knowledge from a well-trained model to estimate spatially varying blur information. It then utilizes all available routing paths, dynamically assigning weights to each expert along every path. As shown in Table 6, each expert is configured with convolutions of varying receptive field sizes, allowing it to identify degraded regions at different scales. To further validate the effectiveness of the DDRE module, we visualize the feature map in Figure 9. Compared to the initial feature map, the DDRE module recovers images with clearer details, such as the phone number on the white car advertisement. Moreover, the different routers exhibit varying abilities to handle blur degradation, demonstrating that our method can adaptively address blur with different degrees of degradation. Given an input feature map $x \in \mathbb{R}^{B \times C \times H \times W}$, the process of router can be defined as::

$$\mathbf{z}_c = \frac{1}{HW} \sum_{i=1}^{H} \sum_{j=1}^{W} x_c(i,j) \tag{13}$$

Then, the pooled vector $\mathbf{z}$ is passed through :

$$\mathbf{u} = \mathbf{W}_2(\sigma(\mathbf{W}_1\mathbf{z} + \mathbf{b}_1)) + \mathbf{b}_2 \tag{14}$$

Finally, the output vector $\mathbf{u}$ is reshaped into $\mathbf{W_j^k} \in \mathbb{R}^{B \times j \times k}$.

## C    DATASETS

We validate the effectiveness of our method using the GoPro dataset Nah et al. (2016), which consists of 2,103 training image pairs and 1,111 evaluation pairs, in line with recent approaches Cui et al. (2024). To evaluate the generalizability of our model, we apply the GoPro-trained model to the HIDE Shen et al. (2019) dataset, containing 2,025 images specifically designed for human-aware

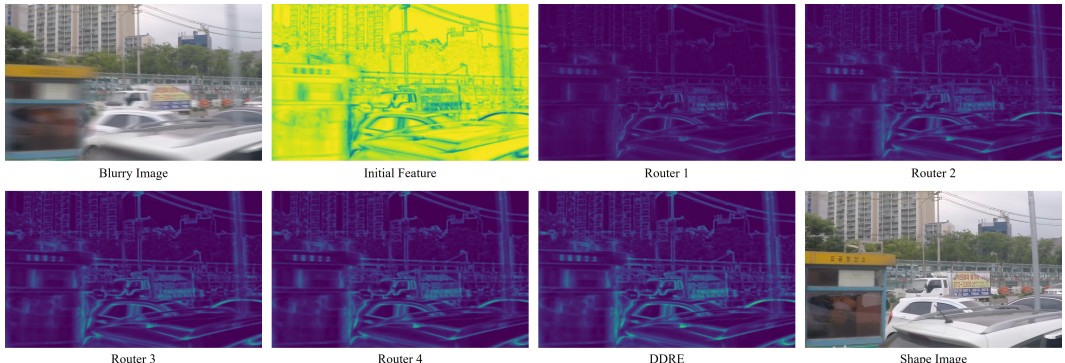

Figure 9: The internal features of DDRE.

motion deblurring. Both the GoPro and HIDE datasets are synthetically generated. Additionally, we assess our method's performance on real-world images using the RealBlur Rim et al. (2020) dataset, which contains 3,758 training image pairs and 980 testing pairs, divided into two subsets: RealBlur-J and RealBlur-R.

## D   MORE PROOF FOR VBLOCK

A Volterra kernel with $L$ terms, it can be expressed as:

$$y_k = h_0 + \sum_{d=1}^{n} \sum_{r_1=0}^{L-1} ... \sum_{r_d=0}^{L-1} h_d(r_1...r_d) \prod_{j=1}^{d} x(k - r_j) \tag{15}$$

where $d$ represents the order, $n$ is the maximum number of $b$, and $r_d$ denotes the memory delay.

For image data, the feature value at location $[x_1, x_2]$ in feature map $F$ is computed using a 2D version of the Volterra kernel, as expressed:

$$
\begin{aligned}
F_z \begin{bmatrix} x_1 \\ x_2 \end{bmatrix} &= V_z(F_{z-1} \begin{bmatrix} x_1 - p_1 : x_1 + p_1 \\ x_2 - p_2 : x_2 + p_2 \end{bmatrix}) \\
&= \sum_{r_{11},r_{21}} h_1 \begin{bmatrix} r_{11} \\ r_{21} \end{bmatrix} x \begin{bmatrix} x_1 - r_{11} \\ x_2 - r_{21} \end{bmatrix} \\
&+ \sum_{\substack{r_{11},r_{21} \\ r_{12},r_{22}}} h_2 \begin{bmatrix} r_{11} \\ r_{21} \end{bmatrix} \begin{bmatrix} r_{12} \\ r_{22} \end{bmatrix} x \begin{bmatrix} x_1 - r_{11} \\ x_2 - r_{21} \end{bmatrix} x \begin{bmatrix} x_1 - r_{12} \\ x_2 - r_{22} \end{bmatrix} \\
&+ ...
\end{aligned}
\tag{16}
$$

where $V_z$ denotes the $z_{th}$ layer of the Volterra kernel, $r_{1i} \in [-p_1, p_1]$ and $r_{2i} \in [-p_2, p_2]$ represent spatial translations in the horizontal and vertical directions, respectively. The complexity of a $n_{-th}$ order Volterra kernel is computed as:

$$\sum_{d=1}^{n} (L[2p_1 + 1][2p_2 + 1])^d \tag{17}$$

The Volterra kernel described earlier is significantly more expressive because it captures higher-order relationships among inputs. However, its computation requires iterated integrals and does not have an efficient GPU implementation. To overcome this limitation, we approximate higher-order behavior by utilizing a convex combination of the first-order and second-order terms of the Volterra kernel.

**Theorem 3.** A cascade of second-order Volterra kernels can approximate the higher-order behavior.

**Proof.** The second-order Volterra kernel can be formulated as:

$$\sigma_{v2}(x) = h_0 + h_1 x + h_2 x^2 \tag{18}$$

After feeding the output of the second-order Volterra kernel $\sigma_{v2}(\cdot)$ back into another second-order Volterra kernel, we obtain an output that reaches up to the fourth order:

$$
\begin{aligned}
\sigma_{v2}(\sigma_{v2}(x)) &= \sigma_{v2}(h_0 + h_1 x + h_2 x^2) \\
&= h_0^{'} + h_1^{'} x + h_2^{'} x^2 + h_3^{'} x^3 + h_4^{'} x^4
\end{aligned}
\tag{19}
$$

where:

$$
\begin{aligned}
h_0^{'} &= h_0 + h_0^2 h_2 \\
h_1^{'} &= h_0 h_1 + 2 h_0 h_1 h_2 \\
h_2^{'} &= h_1^2 h_2 + h_1^2 \\
h_3^{'} &= 2 h_1 h_2 h_2^2 + h_1 h_2 \\
h_4^{'} &= h_2^3
\end{aligned}
\tag{20}
$$

Therefore, we can utilize $K$ instances of the second-order Volterra kernel to implement an $n_{-th}$ order Volterra kernel, where $n = 2^{2^{K-1}}$.

**Theorem 4.** The complexity of an $n_{-th}$ order Volterra kernel, implemented using cascaded second-order Volterra kernels, is calculated as follows:

$$
\sum_{k=1}^{K} [(L_k[2p_{1k}+1][2p_{2k}+1]) + (L_k[2p_{1k}+1][2p_{2k}+1])^2]
\tag{21}
$$

**Proof.** It follows from Eq. 17 that, for a second-order Volterra kernel, the number of parameters required is $[(L_k[2p_1+1][2p_2+1]) + (L_k[2p_1+1][2p_2+1])^2]$. When we utilize $K$ times of the second-order Volterra kernel to implement an $n_{-th}$ order Volterra kernel, it will lead to Eq. 21, which is significantly lower than Eq. 17.

The first order Volterra kernel is similar to the convolutional layer in the conventional CNNs. The second order kernel may be approximated as the concept of separable kernels as :

$$
W_d^2 = \sum_{q=1}^{Q} W_d^{2aq} \otimes W_d^{2bq}
\tag{22}
$$

where $Q$ represents the desired rank of approximation, $W_d^2 \in \mathbb{R}^{(2p_1+1)\times(2p_2+1)\times(2p_1+1)\times(2p_2+1)}$, $W_d^{2aq} \in \mathbb{R}^{(2p_1+1)\times(2p_2+1)\times1}$ and $W_d^{2bq} \in \mathbb{R}^{1\times(2p_1+1)\times(2p_2+1)}$. A larger $Q$ will provide a better approximation of the second order kernel. This is easier to implement with a convolutional library in the first place. Secondly, the complexity is reduced from Eq. 21 to:

$$
\sum_{k=1}^{K} [(L_k[2p_{1k}+1][2p_{2k}+1]) + 2Q(L_k[2p_{1k}+1][2p_{2k}+1])]
\tag{23}
$$

Therefore, we can adjust the value of $Q$ to strike a balance between performance and acceptable computational complexity. We also illustrate the impact of different $Q$ values on the model in the ablation experiments presented in the main text.

## E    LOSS FUNCTION

To optimize the proposed network DHNet by minimizing the following loss function:

$$
\begin{aligned}
L &= L_c(\hat{I}, \overline{I}) + \delta L_e(\hat{I}, \overline{I}) + \lambda L_f(\hat{I}, \overline{I})) \\
L_c &= \sqrt{||\hat{I} - \overline{I}||^2 + \epsilon^2} \\
L_e &= \sqrt{||\triangle\hat{I} - \triangle\overline{I}||^2 + \epsilon^2} \\
L_f &= ||\mathcal{F}(\hat{I}) - \mathcal{F}(\overline{I})||_1
\end{aligned}
\tag{24}
$$

Table 7: Results of alternatives to VBlock.

| Net | MR-VNN Roheda et al. (2024b) | VBlock | PSNR | $\triangle$ PSNR |
|-----|------------------------------|--------|------|------------------|
| (a) | ✔ | | 34.51 | - |
| (b) | | ✔ | 34.75 | +0.24 |

where $\bar{I}$ denotes the target images and $L_c$ is the Charbonnier loss with constant $\epsilon = 0.001$. $L_e$ is the edge loss, where $\triangle$ represents the Laplacian operator. $L_f$ denotes the frequency domains loss, and $\mathcal{F}$ represents fast Fourier transform. To control the relative importance of loss terms, we set the parameters $\lambda = 0.1$ and $\delta = 0.05$ as in Zamir et al. (2021); Cui et al. (2024).

## F MORE ABLATION STUDIES

We provide more ablation studies on the GoPro dataset Nah et al. (2016).

### F.0.1 VBLOCK VS. MR-VNET ROHEDA ET AL. (2024B)

Our work focuses on introducing nonlinear modeling while adaptively handling different degradation regions. To achieve this, we revisit the classical Volterra series **?**, defined as:

$$y_k = h_0 + \sum_{d=1}^{n} \sum_{r_1=0}^{L-1} \cdots \sum_{r_d=0}^{L-1} h_d(r_1, \ldots, r_d) \prod_{j=1}^{d} x(k - r_j), \tag{25}$$

where $d$ denotes the order of nonlinearity, $n$ is the highest considered order, and $r_j$ represents the memory delays. Modern Volterra-based models, such as VNN, VolterraNet, and MR-VNet, can be viewed as deep unfolding implementations of Eq. 25, differing mainly in how and where the Volterra structure is embedded.

MR-VNet Roheda et al. (2024b) treats the Volterra Filter as a fully encapsulated module and directly maps the theoretical Volterra expansion into a block-level implementation without structural refinement. As shown in Eqs. 26–29, it applies a $1 \times 1$ convolution after layer normalization, splits the feature into $2Q$ channel groups,

$$F_{a1}, F_{b1}, \ldots, F_{aQ}, F_{bQ} = Split\big(f_{1 \times 1}(LN(F_{in}))\big), \tag{26}$$

and computes the kernels using $3 \times 3$ convolutions:

$$W_{aq}, W_{bq} = f_{3 \times 3}(F_{aq}), \ f_{3 \times 3}(F_{bq}), \quad q = 1, \ldots, Q. \tag{27}$$

The second-order nonlinear interaction is constructed via outer-product aggregation,

$$W = \sum_{q=1}^{Q} W_{aq} \otimes W_{bq}, \tag{28}$$

and the output is obtained through spatial-channel attention followed by residual fusion:

$$F_{out} = f_{1 \times 1}(SCA(W)) \oplus f_{3 \times 3}(F_{in}). \tag{29}$$

MR-VNet Roheda et al. (2024b) therefore implements the Volterra expansion as a monolithic nonlinear block, without explicitly distinguishing linear terms, nonlinear terms, or the functional role of each component. This limits interpretability and prevents adaptively emphasizing different degradation regions.

Unlike MR-VNet Roheda et al. (2024b), our VBlock incorporates the Volterra mechanism as part of a structured and interpretable computational pipeline. We first generate a refined intermediate representation:

$$F_1 = f_{3 \times 3}\big(f_{1 \times 1}(LN(F_{in}))\big), \tag{30}$$

which enhances the feature basis before nonlinear modeling. The Volterra kernels are computed by splitting a transformed version of $F_1$:

$$W_{a1}, W_{b1}, \ldots, W_{aQ}, W_{bQ} = Split\big(f_{3 \times 3}(F_1)\big). \tag{31}$$

We then explicitly formulate the second-order nonlinear response:

$$X_2 = \sum_{q=1}^{Q} W_{aq} \otimes W_{bq}, \tag{32}$$

and separately compute the linear counterpart:

$$X_1 = f_{3\times3}(F_1). \tag{33}$$

The final output integrates both components through channel attention and retains the input structure via residual fusion:

$$F_{out} = F_{in} \oplus f_{1\times1}(SCA(X_1 \oplus X_2)). \tag{34}$$

From the above formulation, our method provides several inherent benefits:

**1. Explicit decomposition of linear and nonlinear components.** MR-VNet combines all Volterra responses into a single unified block (Eq. 28). In contrast, our method explicitly separates the linear component ($X_1$) and the nonlinear component ($X_2$) (Eqs. 32–33). This design not only preserves essential structural information but also enables controllable nonlinear enhancement and offers clearer interpretability for each transformation stage.

**2. Enhanced feature representation before kernel generation.** While MR-VNet directly constructs Volterra kernels from $f_{1\times1}(LN(F_{in}))$, we introduce a stronger intermediate representation $F_1$ (Eq. 30). This enhanced feature basis yields more expressive second-order interactions, richer Volterra kernels, and better adaptation to spatially heterogeneous degradations.

**3. Deep unfolding structure aligned with optimization principles.** Each step in our VBlock corresponds to a specific functional stage of iterative unfolding:

$$\text{feature refinement} \rightarrow \text{linear term} \rightarrow \text{nonlinear term} \rightarrow \text{joint correction}.$$

Such a design ensures clearer physical meaning and improved optimization stability.

**4. More stable optimization and improved gradient propagation.** The residual formulation in Eq. 34 provides stable gradient flow across iterations. In contrast, MR-VNet's direct aggregation of nonlinear responses (Eq. 29) may amplify noise and lead to less stable training.

**5. Improved handling of spatially varying degradations.** By explicitly modeling $X_1$ (linear) and $X_2$ (nonlinear) and applying attention-based fusion, our method can adaptively focus on severely degraded regions while suppressing nonlinear artifacts in cleaner areas, resulting in more targeted and reliable restoration.

In summary, while MR-VNet uses the Volterra Filter as a single undifferentiated block, our method decomposes the Volterra process into interpretable, structured components. This leads to stronger nonlinear modeling capability, enhanced stability, and better adaptive behavior in spatially varying degradation scenarios.

To validate the advantage of our VBlock over MR-VNet Roheda et al. (2024b), we replace our VBlock with MR-VNet and present the results in Table 1. When replaced, the performance drops by 0.24 dB, demonstrating the superior performance of our VBlock. Additionally, MR-VNet Roheda et al. (2024b) is prone to gradient collapse during practical training.

### F.0.2 RESOURCE EFFICIENT

We evaluate the model complexity of our proposed approach and other state-of-the-art methods in terms of running time and MACs. As shown in Table 8, our method achieves the lowest MACs value while delivering competitive performance in terms of running time.

### F.1 ADDITIONAL VISUAL RESULTS

In this section, we present additional visual results alongside state-of-the-art methods to highlight the effectiveness of our proposed approach, as shown in Figures 10,12,11. It is clear that our model produces more visually appealing outputs for both synthetic and real-world motion deblurring compared to other methods.

Table 8: The evaluation of model computational complexity.

| Method | Time(s) | MACs(G) | Params(M) | PSNR↑ | SSIM↑ |
|---|---|---|---|---|---|
| MPRNet Zamir et al. (2021) | 1.148 | 777 | 20.1 | 32.66 | 0.959 |
| Restormer Zamir et al. (2022) | 1.218 | 140 | 26.2 | 32.92 | 0.961 |
| FSNet Cui et al. (2024) | 0.362 | 111 | 33.29 | 13.3 | 0.963 |
| MR-VNet Roheda et al. (2024b) | 0.388 | 96 | 22.1 | 34.04 | 0.969 |
| AdaRevD-L Mao et al. (2024) | 0.761 | 460 | 210.8 | 34.60 | 0.972 |
| **DHNet(Ours)** | **0.256** | **32** | 30.1 | 33.28 | 0.964 |
| **DHNet-B(Ours)** | 0.499 | 111 | 117.9 | **34.75** | **0.973** |

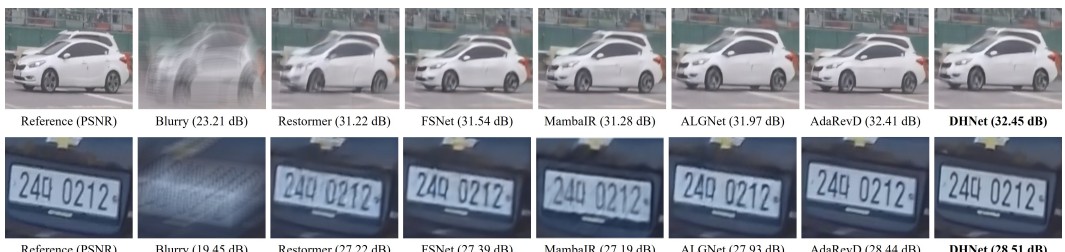

Figure 10: Comparison of image motion deblurring on the GoPro dataset Nah et al. (2016).

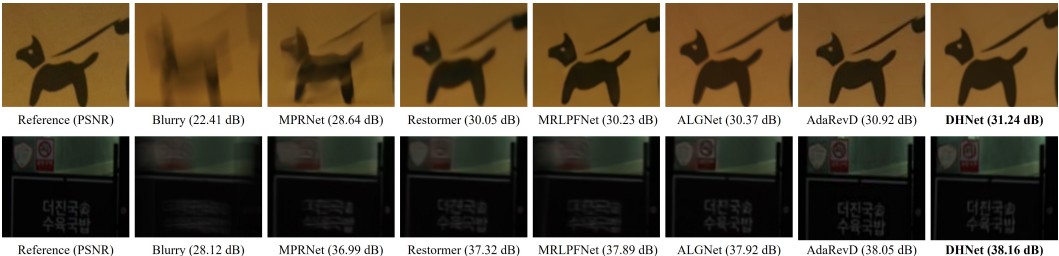

Figure 11: Comparison of image motion deblurring on the RealBlur dataset Rim et al. (2020).

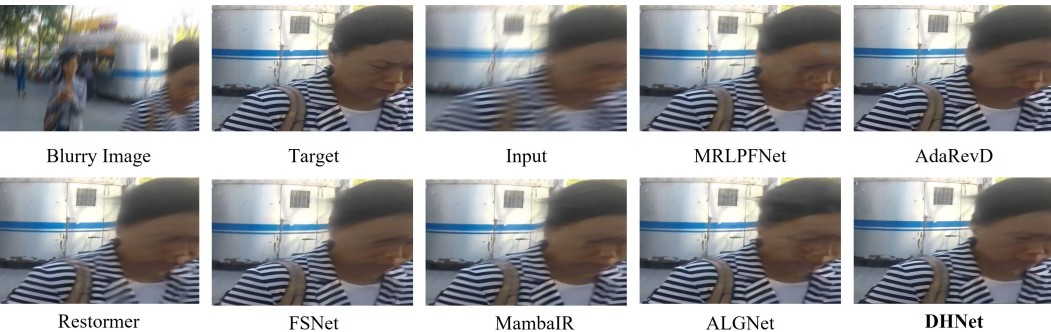

Figure 12: Comparison of image motion deblurring on the HIDE dataset Shen et al. (2019).