# OpenReview forum: "Towards Differential Handling of Various Blur Regions for Accurate Image Deblurring"
_ICLR.cc/2026/Conference — ICLR 2026 Conference Withdrawn Submission_

### Official Review · Reviewer_ZrzN · 2025-10-25

**Soundness:** 2
**Presentation:** 3
**Contribution:** 2
**Rating:** 4
**Confidence:** 5

**Summary:**

This paper proposes a differential handling network (DHNet) for image deblurring. The authors design a Volterra block (VBlock) to integrate the non-linear characteristics into the deblurring network without relying on stacking the number of nonlinear activation functions. In addition, the authors propose the degradation degree recognition expert module (DDRE) that initially incorporates prior knowledge from a pre-trained deblurring network and then utilizes several learnable routers and experts to estimate spatial variable blur information. The proposed DHNet achieves SOTA performance while requiring less computation cost in MACs and inference time compared to the existing methods.

**Strengths:**

1. The proposed DHNet provides an efficient solution for image deblurring. Unlike previous methods that stacke multiple non-liner layers, DHNet utilizes the Volterra block (VBlock) to incorporate non-linear characteristics directly into the network. In addition, DHNet proposes degree recognition expert module (DDRE) that utilizes learnable routers and experts to estimate spatial variable blur information. Experimental results demonstrate that these two components significantly enhance deblurring performance.

2. The proposed DHNet achieve SOTA results on four datasets, including GoPro, HIDE, RealBlur-J, and RealBlur-R, while requiring fewer computational resources in terms of MACs.

**Weaknesses:**

1. The concept of the Volterra block (VBlock) is almost identical to the MR-VNN block in MR-VNet [A]. Furthermore, the introduction section of the MR-VNet paper states, "We also showcase that the recently proposed Non-linear Activation Free Networks (NAFNet) [B] are a special case of the Volterra Formulation." This indicates that MR-VNN has already demonstrated the advantages of the Volterra Formulation over traditional non-linear networks. Consequently, the proposed Volterra block in this paper lacks novelty.

2. Apart from MACs and inference time, the authors should also compare the number of parameters in Table 8.

3. As shown in Table 1, DHNet-B that uses UFPNet significantly outperform DHNet. This demonstrates that the main performance gain is from the pre-trained UFPNet.


[A] MR-VNet: Media Restoration using Volterra Networks. In CVPR 2024.
[B] Simple Baselines for Image Restoration. In ECCV2022.

**Questions:**

1. The proposed Volterra block (VBlock) appears to be identical to the MR-VNN block in MR-VNet [A]. It seems that the authors may have directly adopted the MR-VNN block as the VBlock. Could you provide a detailed clarification on this?

2. Apart from MACs and inference time, the authors should also compare the number of parameters in Table 8.

3. The proposed DHNet-B heavily relies on the pre-trained UFPNet. However, UFPNet is a large model with a substantial number of parameters. Is there any alternative pre-trained model or a more lightweight approach that could be used instead?

[A] MR-VNet: Media Restoration using Volterra Networks. In CVPR 2024.

---

> ### Author Response · Authors · 2025-11-14
>
> ## Dear Reviewer ZrzN,
>
> Thanks for your valuable comments and recognition of our contribution. We highly appreciate your suggestions, and believe they helped improve the quality of our paper. Please find our detailed response below.
>
> &nbsp;
> ## 1. About the detailed clarification on VBlock
>
> We have already addressed this question in **Question 1 to Reviewer UtR9**.
>
> Additionally, in the Appendix "F.0.1," we provide an intuitive analysis by replacing our VBlock with the MR-VNet module. The experimental results show a clear performance drop, further demonstrating that our VBlock not only delivers superior restoration quality but also ensures a more stable training process.
>
> &nbsp;
> ## 2. About the number of parameters in Table 8.
>
> Thank you very much for your suggestions, we have added the model parameters.
> &nbsp;
> | Method                       | Time (s)     | MACs (G) | Params (M) | PSNR ↑    | SSIM ↑    |
> |-----------------------------|--------------|----------|------------|-----------|-----------|
> | MPRNet~\cite{Zamir2021MPRNet}       | 1.148        | 777      | 20.1       | 32.66     | 0.959     |
> | Restormer~\cite{Zamir2021Restormer} | 1.218        | 140      | 26.2       | 32.92     | 0.961     |
> | FSNet~\cite{FSNet}                  | _0.362_      | 111      | 13.3       | 33.29     | 0.963     |
> | MR-VNet~\cite{MR-VNet}              | 0.388        | _96_     | 22.1       | 34.04     | 0.969     |
> | AdaRevD-L~\cite{AdaRevD}            | 0.761        | 460      | 210.8      | _34.60_   | _0.972_   |
> | **DHNet (Ours)**                    | **0.256**    | **32**   | 30.1       | 33.28     | 0.964     |
> | **DHNet-B (Ours)**                  | 0.499        | 111      | 117.9      | **34.75** | **0.973** |
>
> &nbsp;
> ## 3. About the alternative pre-trained model
> This pre-trained model can be substituted with any defuzzification network. Its primary purpose is to learn degradation patterns, which in turn guide the decoder to handle varying degradation levels across different regions.

---

### Official Review · Reviewer_Ypjw · 2025-10-26

**Soundness:** 3
**Presentation:** 2
**Contribution:** 2
**Rating:** 4
**Confidence:** 5

**Summary:**

The paper “MBMamba: When Memory Buffer Meets Mamba for Structure-Aware Image Deblurring” aims to enhance the spatial structure modeling capability of Mamba-based image deblurring frameworks. The authors point out that the traditional Mamba’s “flatten-and-scan” strategy leads to local pixel forgetting and channel redundancy, thereby weakening spatial feature aggregation. To address this issue, they propose the MBMamba framework, whose core component is the Memory Buffering Mechanism (MemVSSM). This module preserves historical feature information through a memory buffer and fuses it via a Feature Cross-Attention Module (FCAM), strengthening local context modeling. In addition, an Ising-inspired regularization loss is designed to simulate pixel-level “mutual attraction” energy minimization, thereby maintaining structural consistency.

**Strengths:**

The MemVSSM module is designed to enhance local context preservation via feature partitioning and temporal buffering. The physically inspired Ising Loss is incorporated into the deblurring loss function, enforcing spatial coherence through energy minimization. Notably, this module extends the spatial-awareness capability of the Mamba backbone without altering its main architecture, achieving both efficiency and accuracy.

The proposed method provides a new perspective on memory-based state-space modeling in visual tasks. It achieves superior results across four major benchmarks — GoPro, HIDE, RealBlur-R, and RealBlur-J — while maintaining significantly lower FLOPs and inference latency compared to Transformer-based methods.

**Weaknesses:**

Furthermore, the title and abstract also stress “Various Blur Regions”, but the experiments do not demonstrate how this claim is achieved. There is no explanation of how the network distinguishes between different blur regions, nor how it dynamically adapts processing across regions with varying blur intensities.

Although VBlock refines the Volterra structure from MR-VNet, the core idea of using higher-order kernels to model nonlinearities is not novel. The paper does not sufficiently explain why DHNet achieves better stability or generalization compared to prior Volterra-based networks.

While the paper presents multiple theorems (Theorem 1–4) demonstrating that the Volterra kernel can approximate any continuous nonlinear function, it lacks experimental or visual evidence quantifying these properties—for instance, comparisons of Volterra versus ReLU in terms of convergence speed or stability. As a result, the theoretical claims are not tightly connected to empirical performance.

The DDRE module involves multiple routers and experts, yet only a “single-router case” is visualized. The overall routing mechanism and computational process are not clearly illustrated. Moreover, the weight assignment among experts lacks interpretability and visualization, making the internal logic of this module difficult to fully grasp.

Finally, the paper mainly compares DHNet with CNN-based and a few Transformer-based methods, but does not discuss whether DHNet can truly replace Transformers in capturing global dependencies. For instance, no visualization of global modeling capability is provided. Since DHNet remains primarily convolution-based, its advantage in handling complex dynamic blur still requires further empirical verification.

**Questions:**

See weakness

---

> ### Author Response · Authors · 2025-11-14
>
> ## Dear Reviewer Ypjw,
> Thank you very much for your review. Your comments are of great help to the improvement of our paper. Next, we will answer your questions one by one.
>
> &nbsp;
> ## 1. About the dynamically adapts processing across regions with varying blur intensities
>
> To adaptively identify degradation degrees across varying sizes of blur regions, we propose a degradation degree recognition expert module (DDRE). DDRE first integrates prior knowledge from a well-trained model to estimate the spatially variable blur information. This enables the router to map the learned degradation representation and assign weights to experts based on both the degree of degradation and the size of the regions.
>
> Unlike conventional mixture of experts methods, DDRE utilizes all available routing paths, dynamically assigning weights to each expert along every path. As shown in Table 6, each expert is configured with convolutions of varying receptive field sizes, allowing it to identify degraded regions at different scales.
>
>
> To further demonstrate the effectiveness of our DDRE module, we visualize the features of a routing path weighted over all experts, as shown in Figure 6. It is clear that each expert handles the inconsistent degradation region as well as  the degree.
> We also visualize the feature map in Figure 9. Compared to the initial feature map, the DDRE module recovers images with clearer details, such as the phone number on the white car advertisement. Moreover, the different routers exhibit varying abilities to handle blur degradation, demonstrating that our method can adaptively address blur with different degrees of degradation.
>
>
>
>
>
> &nbsp;
> ## 2. About the VBlock
> We have already addressed this question in **Question 1 to Reviewer UtR9**.
>
> Additionally, in the Appendix "F.0.1," we provide an intuitive analysis by replacing our VBlock with the MR-VNet module. The experimental results show a clear performance drop, further demonstrating that our VBlock not only delivers superior restoration quality but also ensures a more stable training process.
>
> &nbsp;
> ## 3. About the  convergence speed or stability
> It is well-known that nonlinear activation functions introduce computational overhead, so we did not provide additional proof. Following your suggestion, we replaced the VBlock function with ReLU. The experimental results indicate that while the MACs increased from 32 to 34, the performance decreased from 33.28 to 32.96.
>
> &nbsp;
> ## 4. About the  overall routing mechanism and computational process
> Although Figure 3 visualizes only the single-router case for simplicity, the proposed DDRE module indeed supports multiple routers, each dynamically assigning weights to all experts, rather than selecting a single expert or a sparse subset as in traditional MoE frameworks. The complete routing mechanism  process is summarized as follows.
>
> Given an input feature map $x \in \mathbb{R}^{B \times C \times H \times W}$, the process of router can be defined as:
>
> $\mathbf{z}_c = \frac{1}{H W} \sum _{i=1}^{H} \sum _{j=1}^{W} x _{c}(i,j) $
>
> Then, the pooled vector $\mathbf{z}$ is passed through :
>
> $\mathbf{u} = \mathbf{W}_2 (\sigma\left( \mathbf{W}_1 \mathbf{z} + \mathbf{b}_1 \right)) + \mathbf{b}_2$
>
> Finally, the output vector $\mathbf{u}$ is reshaped into
> $\mathbf{W_j^k} \in \mathbb{R}^{B \times j \times k}.$
>
>
> &nbsp;
> ## 5. About the  global modeling capability
> In this work, our primary objective is to address two key issues: (1) existing methods rely on stacking numerous nonlinear activation functions to approximate nonlinear properties, and (2) they often neglect the varying degrees of degradation across different blurred regions of an image. Accordingly, we did not focus on modifying the model architecture extensively, instead adopting the CNN-based NAFNet as our baseline. While our approach may not capture global dependencies as effectively as Transformer-based models, it offers significantly lower computational complexity. Furthermore, our VBlock can be integrated into existing Transformer-based restoration models, replacing traditional activation functions to enhance their performance.

---

> ### Comment · Reviewer_Ypjw · 2025-11-27
> **Relpy**
>
> First, the table only shows that your experts use convolutional kernels of different sizes; it does not demonstrate that they have learned any knowledge capable of distinguishing blurred from non-blurred regions. Moreover, there is no experimental evidence proving that these experts adopt targeted processing strategies for different levels of blur.
>
> Second, in Figures 6 and 9, I cannot see that DDRE successfully identifies degraded regions. For example, the buildings in the background are relatively clear, whereas the moving car is relatively blurred. However, the visualized maps assign similar colormap values to these two areas. Therefore, I do not believe DDRE is actually recognizing degraded regions. Based on the current experimental results, the motivation of the paper does not hold.
>
> I hope the author can clarify my above concerns in detail.

---

> > ### Author Response · Authors · 2025-11-28
> >
> > We appreciate the reviewer’s careful examination of the visualizations.  We would like to clarify that the issue stems from the visualization method rather than the underlying routing behavior.
> >
> > First, in Figure 6, the colormap was normalized independently for each image, which can make regions with different absolute blur levels appear visually similar.  Nevertheless, a closer inspection still reveals clear differences: for example, the area highlighted by the red box exhibits notably stronger activation than the background buildings, and this contrast is especially evident in the five Korean characters along the left border.  These cues indicate that DDRE does respond more strongly to heavily degraded regions.
> >
> > Second, regarding Figure 9, the purpose of the visualization is not to compare color intensity within each map but to demonstrate the inconsistency across different routes.  Each route produces distinct activation patterns, reflecting the diverse weighting assigned to experts under different degradation conditions.
> >
> > With these visualizations in mind, we can further clarify how DDRE adaptively handles varying levels of blur.  Because each expert possesses a different receptive field, they inherently focus on regions of different spatial scales and aggregate information differently.  This structural diversity is fundamental to DDRE’s ability to address blur patterns of varying sizes.  The visualizations in Figure 6 support this behavior: each expert responds differently to the same degraded area (e.g., the telephone pole).  Expert 1 shows noticeably stronger activation around the red-box text compared with Expert 2, demonstrating that experts specialize in different types or strengths of degradation.
> >
> > Finally, Figure 9 illustrates the per-route weight coefficients (before softmax), which serve as expert-specific weighting factors.  These route-dependent differences indicate that DDRE assigns distinct weights to experts under different degradation patterns.  Combined with the aggregation in Equation (11), this mechanism allows DDRE to implement differential processing for regions with varying degrees of blur.

---

### Official Review · Reviewer_UtR9 · 2025-10-31

**Soundness:** 2
**Presentation:** 3
**Contribution:** 1
**Rating:** 2
**Confidence:** 4

**Summary:**

I have reviewed this paper in a previous venue. The submission appears to be largely identical to that version. Below, I include my previous review (with possible minor edits for clarity).

This paper proposes DHNet, a new image deblurring model that consists of 1) the VBlock module for incorporating nonlinear characteristics and 2) the DDRE module for dealing with varying degradation degrees in blurred images. The deblurring performance is evaluated across several image deblurring benchmark datasets.

**Strengths:**

- Experimental results demonstrate that DHNet achieves the SOTA performance with higher efficiency. The deblurring results, both visualization and PSNR/SSIM metrics on commonly used benchmarks like GOPRO and HIDE, are convincing.
- The paper is well-written and easy to follow.

**Weaknesses:**

- The novelty is limited. 1) The VBlock is borrowed from previous research[1]. Meanwhile, it has already been applied to the image deblurring task by [2]. Clarifying the novelty of VBlock compared with these works is expected. 2) Dealing with varying degradation degrees in image deblurring has been discussed by prior works like [3]. However, these works and the differences are not discussed in the paper.
- In Fig. 2, the x-axis lacks necessary labels, making it difficult to assess the improvement in computational efficiency of the model.
- The ablation study and analysis of the DDRE module are insufficient. Fig. 6 does not convincingly demonstrate that the Experts focus on regions with different degradation degrees (since visualizations of different channels from the output features of a single network layer could also produce similar effects). I recommend that the authors adopt more rigorous quantitative metrics to verify that the focusing regions of different Experts indeed correspond to different degradation degrees.

[1] VolterraNet: A Higher Order Convolutional Network With Group Equivariance for Homogeneous Manifolds. TPAMI 2022.

[2] MR-VNet: Media Restoration using Volterra Networks. CVPR 2024.

[3] Motion deblurring algorithm for wind power inspection images based on Ghostnet and SE attention mechanism. TIP 2023.

**Questions:**

Please refer to the weaknesses.

---

> ### Author Response · Authors · 2025-11-14
>
> ## Dear Reviewer UtR9,
>
>
> Many thanks for your valuable time and comments. We carefully address your concerns in detail.
>
> ## 1. About the novelty of VBlock
> In the Appendix “F.0.1”, we provide an intuitive analysis by replacing our VBlock with the MR-VNet module. The experimental results demonstrate a clear performance drop, further indicating that our VBlock not only achieves better restoration quality but also exhibits a more stable training process.
>
> Our work focuses on introducing nonlinear modeling while adaptively handling different degradation regions. To achieve this, we revisit the classical Volterra series, defined as:
>
> $y_k = h_0 + \sum_{d=1}^{n} \sum_{r_1=0}^{L-1} \cdots \sum_{r_d=0}^{L-1} h_d(r_1,\ldots,r_d)\prod_{j=1}^{d}x(k-r_j) \quad(1)$
>
> Modern Volterra-based models, such as VNN, VolterraNet, and MR-VNet, can be viewed as deep unfolding implementations of Eq.(1), differing mainly in how and where the Volterra structure is embedded.
>
> MR-VNet treats the Volterra Filter as a fully encapsulated module and directly maps the theoretical Volterra expansion into a block-level implementation without structural refinement. As shown in Eqs.(2-5), it applies a $1\times1$ convolution after layer normalization, splits the feature into $2Q$ channel groups:
>
> $F_{a1},F_{b1},\ldots,F_{aQ},F_{bQ}=Split\big(f_{1\times1}(LN(F_{in}))\big) \quad (2)$
>
> and computes the kernels using $3\times3$ convolutions:
>
> $W_{aq},W_{bq}=f_{3\times3}(F_{aq}),\ f_{3\times3}(F_{bq})\quad q=1,\ldots,Q. \quad (3)$
>
> The second-order nonlinear interaction is constructed via outer-product aggregation:
>
> $W=\sum_{q=1}^{Q} W_{aq}\otimes W_{bq} \quad (4)$
>
> and the output is obtained through spatial-channel attention followed by residual fusion:
>
> $F_{out}=f_{1\times1}(SCA(W))\oplus f_{3\times3}(F_{in}) \quad (5)$
>
> MR-VNet therefore implements the Volterra expansion as a monolithic nonlinear block, without explicitly distinguishing linear terms, nonlinear terms, or the functional role of each component. This limits interpretability and prevents adaptively emphasizing different degradation regions.
>
> Unlike MR-VNet, our VBlock incorporates the Volterra mechanism as part of a structured and interpretable computational pipeline. We first generate a refined intermediate representation:
>
> $F_1 = f_{3\times3}(f_{1\times1}(LN(F_{in}))) \quad (6) $
>
> which enhances the feature basis before nonlinear modeling. The Volterra kernels are computed by splitting a transformed version of $F_1$:
>
> $W_{a1},W_{b1},\ldots,W_{aQ},W_{bQ}=Split\big(f_{3\times3}(F_1)\big) \quad (7)$
>
> We then explicitly formulate the second-order nonlinear response:
>
> $X_2=\sum_{q=1}^{Q} W_{aq}\otimes W_{bq} \quad (8)$
>
> and separately compute the linear counterpart:
>
> $X_1=f_{3\times3}(F_1) \quad (9)$
>
> The final output integrates both components through channel attention and retains the input structure via residual fusion:
>
> $F_{out}=F_{in}\oplus f_{1\times1}(SCA(X_1\oplus X_2)) \quad(10) $
>
> From the above formulation, our method provides several inherent benefits:
>
> **Explicit decomposition of linear and nonlinear components.**
> MR-VNet combines all Volterra responses into a single unified block (Eq.(4)). In contrast, our method explicitly separates the linear component ($X_1$) and the nonlinear component ($X_2$) (Eqs.(8-9)). This design not only preserves essential structural information but also enables controllable nonlinear enhancement and offers clearer interpretability for each transformation stage.
>
> **Enhanced feature representation before kernel generation.**
> While MR-VNet directly constructs Volterra kernels from $f_{1\times1}(LN(F_{in}))$, we introduce a stronger intermediate representation $F_1$ (Eq.(6)). This enhanced feature basis yields more expressive second-order interactions, richer Volterra kernels, and better adaptation to spatially heterogeneous degradations.
>
> **Deep unfolding structure aligned with optimization principles.**
> Each step in our VBlock corresponds to a specific functional stage of iterative unfolding: feature refinement $\rightarrow $ linear term $ \rightarrow $nonlinear term $ \rightarrow $joint correction. Such a design ensures clearer physical meaning and improved optimization stability.
>
> **More stable optimization and improved gradient propagation.**
> The residual formulation in Eq.(10) provides stable gradient flow across iterations. In contrast, MR-VNet’s direct aggregation of nonlinear responses (Eq.(5)) may amplify noise and lead to less stable training.
>
> **Improved handling of spatially varying degradations.**
> By explicitly modeling $X_1$ (linear) and $X_2$ (nonlinear) and applying attention-based fusion, our method can adaptively focus on severely degraded regions while suppressing nonlinear artifacts in cleaner areas, resulting in more targeted and reliable restoration.

---

> ### Author Response · Authors · 2025-11-14
>
> ## 2. Regarding the novelty in handling varying degrees of degradation
> Based on your question, reference [3] is titled “Motion deblurring algorithm for wind power inspection images based on GhostNet and SE attention mechanism”, supposedly published in TIP 2023.
> However, we did not find this article in IEEE TIP. Instead, the actual publication appears in IET Image Processing:
>
> #### Gao, R., Wang, T.:  Motion deblurring algorithm for wind power inspection images based on Ghostnet and SE attention mechanism.  IET Image  Process. 17, 291-300 (2023). https://doi.org/10.1049/ipr2.12634
>
> In this paper, the main contribution is the construction of a new dataset that captures local blur variations in wind power inspection images, rather than proposing an algorithm specifically tailored to these blur differences.
>
> Therefore, its focus is fundamentally different from ours. **Our work emphasizes algorithmic design** that adaptively handles different regions and varying blur levels, whereas **their work focuses on building a blur dataset**  reflecting real-world wind power inspection conditions.
>
>
> &nbsp;
> ## 3. Regarding the x-axis lacks necessary labels
> For the x-axis coordinates, we set them to 30, 40, 50, …, 100, 200, 300, …, and have updated Figure 2 accordingly.
>
> &nbsp;
> ## 4. Regarding the ablation study and analysis of the DDRE module
>
> At the very beginning of our experiments, and also considering your observation, we noted that different channels from the output features of a single network layer could potentially produce similar effects. To address this, we conducted a visual comparison across all corresponding channels of each expert. The results indicate that the attention regions of each expert are indeed inconsistent.
>
> In Figure 6, to provide a fair and intuitive demonstration of DDRE’s effectiveness, we averaged all channels of each expert feature map before visualization. Additionally, further experiments reported in the Appendix (Figure 9) show that, compared to the initial feature map, the DDRE module restores images with clearer details—for example, the phone number on the white car advertisement.
> Moreover, the different routers exhibit varying capabilities in handling blur degradation, demonstrating that our method can adaptively address blur of different intensities.
>
> Finally, since each expert in DDRE uses convolutions with different receptive field sizes, the effective attention areas differ theoretically, which further validates the correctness of our results.

---

### Note · Authors · 2026-01-26

I have read and agree with the venue's withdrawal policy on behalf of myself and my co-authors.

---

### Meta-Review · Area_Chair_daD7 · 2025-12-25

**Summary:**

The submission reports strong quantitative results and efficiency on standard deblurring benchmarks, and the paper is generally readable. However, multiple reviewers raised substantial concerns about (i) limited novelty of the core VBlock design relative to prior Volterra-based restoration—suggesting the main contribution may largely be a re-use/refinement of existing modules—and (ii) insufficiently convincing evidence that the proposed “differential handling” mechanism (DDRE) truly identifies and treats different blur regions as claimed. In addition, there is concern that a significant portion of the reported gains may be attributable to reliance on a strong pre-trained deblurring model rather than the proposed new components.

Given these unresolved issues around novelty, causal attribution, and the central mechanism’s supporting evidence, I regretfully recommend rejection.

**Reviewer Concerns:**

**Concerns addressed by rebuttal / discussion:**
- **Clarity/format fixes:** One reviewer noted missing axis labels in an efficiency figure; the authors state this was corrected in the revision.
- **Reporting completeness:** A reviewer requested parameter-count comparisons in addition to MACs/time; the authors report adding these numbers.
- **Implementation-level clarifications:** The authors provided additional explanation and replacement experiments to argue that their VBlock is not a direct copy of MR-VNet, and clarified details of their DDRE visualizations (e.g., normalization choices).

**Remaining / partially outstanding concerns (key drivers for rejection):**
- **Novelty and positioning remain unconvincing:** Reviewers maintain that VBlock is conceptually very close to prior Volterra/activation-free restoration blocks, including MR-VNet-style designs; the incremental nature of the core architectural idea remains a primary concern.
- **Attribution of gains / dependence on pre-trained model:** A reviewer highlighted that a stronger pre-trained backbone variant (DHNet-B with a pre-trained model) appears to account for a large portion of performance improvements, weakening the causal link to the proposed contributions.
- **“Differential handling” evidence not sufficiently compelling:** Multiple reviewers questioned whether DDRE genuinely distinguishes blur vs. non-blur regions and whether experts adopt targeted strategies for different degradation levels. Even after author clarification, at least one reviewer explicitly remained unconvinced and stated that the paper’s motivation is not supported by the current evidence.
- **Interpretability / mechanism clarity:** Concerns remain about the routing mechanism visualization and interpretability of expert weights, as well as the lack of stronger quantitative validation of “region-wise specialization.”
- **Global modeling discussion remains weak:** A reviewer noted limited evidence that this primarily convolutional approach can handle complex/global blur dynamics comparably to transformer-style global modeling, and requested stronger supporting evidence/visualization.

**Reviewer Scores:**

- **UtR9:** **2 (reject)**. Given the emphasis on limited novelty and insufficient DDRE analysis, I expect the score to remain **2**.
- **Ypjw:** **4 (marginally below threshold)**. The reviewer’s follow-up comments indicate continued skepticism about DDRE and the core motivation; I expect it to remain **~4** (possibly lower, but conservatively unchanged).
- **ZrzN:** **4 (marginally below threshold)**. Novelty and “gain attribution to pre-training” concerns are central; I expect it to remain **~4**.

---

### Decision · Program_Chairs · 2026-01-26

Reject